# Contribution of irreversible non-180° domain to performance for multiphase coexisted potassium sodium niobate ceramics

Bo Wu [1,2,3], Lin Zhao[1,3], Jiaqing Feng[1,3], Yiting Zhang[1,3], Xilong Song[1,3], Jian Ma[1,3], Hong Tao[1,3] ✉, Ze Xu [2], Yi-Xuan Liu[2], Shidong Wang [4] ✉, Jingtong Lu[2], Fangyuan Zhu[5], Bing Han [6] ✉ & Ke Wang [2] ✉

Despite the dominance of lead-based piezoelectric materials with ultrahigh electric-field-induced strain in actuating applications, seeking eco-friendly substitutes with an equivalent performance remains an urgent demand. Here, a strategy of regulating the irreversible non-180° domain via phase engineering is introduced to optimize the available strain (the difference between the maximum strain and the remnant strain in a unipolar strain curve) in the lead-free potassium–sodium niobate-based piezoelectric ceramics. In situ synchrotron X-ray diffraction and Rayleigh analysis reveal the contribution of the non-180° domain to available strain in the tetragonal–orthorhombic–rhombohedral phase boundary. The reducing orthorhombic phase and increasing rhombohedral/tetragonal phase accompanied by the reduced irreversible non-180° domain are obtained with increasing doping of $Sb^{5+}$, resulting in an enlarged available strain due to the significantly lowered remnant strain. This optimization is mainly attributed to the reduced irreversible non-180° domain wall motion and the increased lattice distortion, which are beneficial to decrease extrinsic contribution and enhance intrinsic contribution. The mesoscopic structure of miniaturized nanosized domain with facilitated domain switching also contributes to the enhancement of available strain due to the improved random field and decreased energy barrier. The study will shed light on the design of lead-free high-performance piezoelectric ceramics for actuator applications.

Piezoelectric materials with the unique functionality of interconverting mechanical and electric energies, play a significant role in applications of many aspects, such as medicine, electronics, military, etc.[1–9]. The market demand for piezoelectric applications is expected to grow to a multi-billion-dollar with an annual growth of 13.2%[10]. As the most representative, piezoelectric actuators which can convert the electrical signal to displacement precisely, are expected to grow to 35.4 billion USD by 2026[11–22]. For several decades, lead zirconate titanate [Pb(Zr, Ti)O$_3$, PZT]-based piezoelectric actuators have dominated the market because of their outstanding performance[11]. Recently, restrictions on

the use of lead in electronic devices have gradually come into practice due to ever-increasing ecological and environmental concerns, inducing great attention to lead-free piezoelectric materials[8–10].

Electric field-induced-strain (or electrostrain) performance is one of the most important figure-of-merits for piezoelectric actuators[12,13]. Many breakthroughs have been made via microstructure manipulation in lead-free ceramics, including phase engineering, defect engineering, etc.[15–22]. For example, the giant electrostrain values of 0.7% ($d_{33}^*$ ~1400 pm/V) and 1.05% ($d_{33}^*$ ~2100 pm/V) have been achieved in bismuth sodium titanate [(Bi, Na)TiO$_3$, BNT] and potassium–sodium niobate [(K, Na)NbO$_3$, KNN]-based ceramics due to the electric field-induced phase transition and the interaction between defect dipoles and domain switching, respectively, leading to a new wave of research on this topic[15,17]. However, there remain critical challenges for practical applications, such as hysteresis and nonlinearity of electrostrain[11–13]. Additionally, a crucial issue of lead-free piezoelectric actuators is to obtain large available strain performance that is comparable to the lead-based ones. Generally, the actuating performance is assessed merely by the difference between poling strain ($S_{pol}$) and remanent strain ($S_{rem}$) (See Fig. 1a), which can be estimated via the following equation[12]:

$$d_{33}^* = \frac{S_{pol} - S_{rem}}{E_{max}} \tag{1}$$

where the $E_{max}$ is the maximum electric field and $d_{33}^*$ is the converse piezoelectric coefficient. Obviously, the higher $S_{pol}$ and the lower $S_{rem}$, the better actuating performance. Namely, the actuating performance can be modified by enhancing $S_{pol}$ and/or restricting $S_{rem}$. $S_{pol}$ can originate from multiple mechanisms, including the converse piezoelectric effect, non-180° domain wall motion, electrostriction, and possible electric field-induced phase transition[11–13]. On the other hand, the $S_{rem}$ is mainly generated by irreversible non-180° domain evolution which can keep the orientation after withdrawing the electric field[12]. Compared to the complex mechanisms for $S_{pol}$, $S_{rem}$ with a rather straightforward contribution of the irreversible non-180° domain might be easier to regulate. Modifying the irreversible non-180° domain could be an effective approach to optimize available strain.

It is known that a change in lattice symmetry can lead to a variation in domain morphology, which can be accompanied by phase engineering and defect engineering[23–30]. Among them, phase engineering has been demonstrated to be the most promising approach[3]. KNN-based lead-free ceramics with multiphase boundary, which exhibit excellent piezoelectric response, are a great candidate for piezoelectric applications but there are challenges ahead[27]. Of the utmost importance, the

mechanism for $S_{rem}$ needs to be urgently revealed, especially the connection between available strain ($S_{uni}$–$S_{rem}$) and irreversible non-180° domain for improving the practical actuating strain[12]. Here, the contribution of the irreversible non-180° domain to available strain is revealed via adjusting the rhombohedral–orthorhombic–tetragonal (R–O–T) phase boundary in high-performance KNN-based ceramics (Fig. 1b). Significantly, a high available strain is realized with reduced irreversible non-180° domain and decreased domain size based on increased R/T and decreased O phase. The underlying mechanism of enhancement is discussed based on the reduced extrinsic contribution of domain evolution and elevated intrinsic contribution of phase transition, accompanied by decreased domain size and facilitated domain switching. This work demonstrates an effective approach to the design of high-strain performance piezoelectric materials for actuators.

## Results

### Phase structure and macro-performance

A typical perovskite structure is obtained without any impurity phase in all the ceramics (Supplementary Fig. 1a, b)[3]. Room temperature phase transitions including $T_{R–O}$ and $T_{O–T}$ are observed for $x = 0.035$–0.055, indicating the R–O–T phase boundary (Supplementary Figs. 1c, 2 and Supplementary Table 1) which is verified by the Rietveld refinement (Fig. 2a, Supplementary Fig. 3, and Supplementary Table 2). The modified phase structure and pronounced diffusion behavior in phase transition are achieved by the increasing $T_{R–O}$ and decreasing $T_{O–T}$ as well as the enlarged compositional/structural heterogeneity (Supplementary Table 1 and Supplementary Fig. 4). The dominant R and O phases coexist with little T phase because of the much higher $T_{O–T}$ for $x = 0.035$, while the R–O–T phase boundary with strongly diffused behavior is obtained for $x = 0.055$ along with obviously diffused XRD and dielectric peaks. Samples with appropriate R–O–T phase structure ($x = 0.04$–0.05) are adopted to explore the effect on domain evolution from phase content. Significantly, the R and T phases increase as the O phase decreases with increasing Sb$^{5+}$ content (Fig. 2b). It is known that the R phase contains spontaneous polarizations along 8 directions of <111>, which leads to the formation of 0°, 71°, 90°, 109°, and 180° domain walls[31]. For the O phase, there are 12 directions along <011> which leads to the formation of 0°, 60°, 90°, 120°, and 180° domain walls, while for the T phase, there are 6 directions along <001>, which leads to the formation of 0°, 90°, and 180° domain walls, respectively (Fig. 2c–e)[31]. The R–O–T phase boundary gradually changes to the O–T phase, T phase, and cubic phase finally with increasing temperature, leading to the reduced non-180° domain and weakened domain response (Fig. 2f)[3]. Thus, the domain characteristics depend on phase content correspondingly. Especially, the

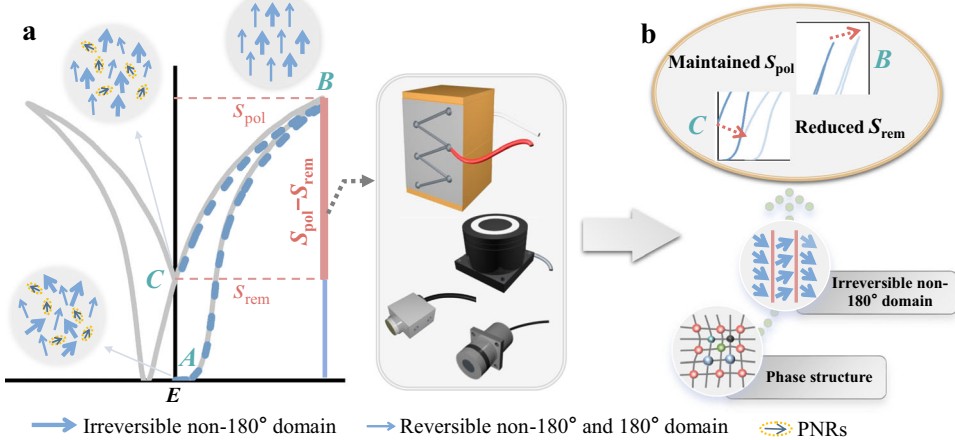

**Fig. 1 | Design strategy for large available strain. a** Schematic diagram of strain vs domain evolution under applied electric field. **b** Schematic diagram of the relationship between available strain ($S_{pol}$–$S_{rem}$) and irreversible non-180° domain.

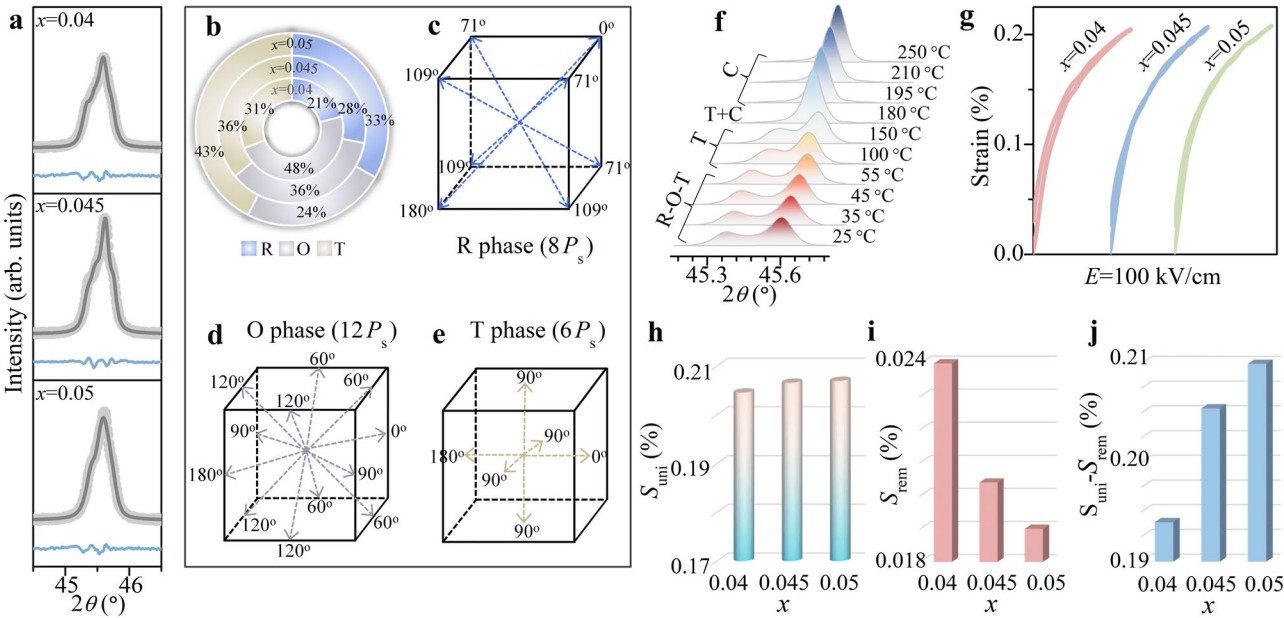

**Fig. 2 | Phase structure and strain performance. a** Rietveld refinement of the ceramics with $x = 0.04$, $x = 0.045$, and $x = 0.05$. **b** Phase fraction. **c** Rhombohedral (R) phase polarization direction. **d** Orthorhombic (O) phase polarization direction. **e** Tetragonal (T) phase polarization direction. **f** Temperature-dependent X-ray diffraction of $x = 0.045$. Evolution of **g** unipolar strain, **h** maximum unipolar strain ($S_{uni}$), **i** remnant strain ($S_{rem}$), and **j** available strain ($S_{uni}-S_{rem}$).

variation of the non-180° domain could yield different performances (Supplementary Table 3 and Fig. 5). Improved piezoelectric and ferroelectric properties are achieved for $x = 0.045$. Meanwhile, strain evolution is further emphasized since it is a figure of merit for piezoelectric actuators. From the unipolar strain curves (Fig. 2g), $S_{rem}$ drops from 0.024% to 0.019% when $x$ increases from 0.04 to 0.05 along with little changed $S_{uni}$ (-0.206%), contributing to enhanced $S_{uni}-S_{rem}$, where the largest value of 0.187% is observed for $x = 0.05$ (Fig. 2h, i). The high available strain achieved is favorable for the piezoelectric actuating applications[12,13].

### Characterization of irreversible non-180° domain

The electrostrain originates from both intrinsic lattice strain and extrinsic strain from the non-180° domain, where the latter often plays a major role[12,21]. Thus, the evolution of domains upon applied electric field is explored. In situ, X-ray diffraction (XRD) has been recognized as a powerful approach to structure analysis during poling for piezoelectric materials[32–44]. Notably, the shift of diffraction peak position and variation of peak intensity can reflect the piezoelectric lattice strain due to the change of the unit cell and non-180° domain switching, respectively[34]. Of particular interest, the XRD patterns of (111) and (200) reflections are specifically investigated[34,43]. The domain evolution under different electric fields [virgin: 0 kV/mm→in situ: $3E_c$ (coercive electric field)→poled: 0 kV/mm] is revealed with well-fitted diffraction with R, O, and T phases (Fig. 3a and Supplementary Fig. 6). The volume fractions ($F$) of phases within the R−O−T multiphase coexistence can be approximately derived according to the following formula[45]:

$$F_R = \frac{SI_R}{SI_T + SI_O + SI_R} \tag{2}$$

$$F_O = \frac{SI_O}{SI_T + SI_O + SI_R} \tag{3}$$

$$F_T = \frac{SI_T}{SI_T + SI_O + SI_R} \tag{4}$$

where $SI_R$, $SI_R$, and $SI_R$ are the sums of peak intensities of (200) reflections for R, O, and T phases, respectively. The volume fractions of the phases are obtained from the characterization of the virgin state. Increasing R and T phases, as well as the decreasing O phase, are revealed due to increasing $Sb^{5+}$ content (Supplementary Fig. 7), which conforms to the result from the Rietveld refinement of XRD patterns. Importantly, a considerably enhanced volume fraction of non-180° domains parallel to the electric field direction is observed during the application of external electric field in all samples, which is evidenced by the changed peak intensity ratios between the $111_R/11\text{-}1_R$, $202_O/020_O$, and $002_T/200_T$ reflections for the R, O, and T phases, respectively[34,43]. The volume fraction ($f$) of domains parallel to the electric field direction for R, O, and T phases can be expressed by the following equation[34]:

$$f_R = \frac{I_{(111)R}}{I_{(111)R} + I_{(11-1)R}} \tag{5}$$

$$f_O = \frac{I_{(202)O}}{I_{(202)O} + I_{(020)O}} \tag{6}$$

$$f_T = \frac{I_{(002)T}}{I_{(002)T} + I_{(200)T}} \tag{7}$$

The $f_R$, $f_O$, and $f_T$ first increase and then decrease in the ceramics within the applied electric field process from 0 kV/mm to $3E_c$, and subsequently to 0 kV/mm (Supplementary Fig. 8), which can be ascribed to the reorientation of the non-180° domain under electric field. Significantly, the volume fraction of domains parallel to the electric field direction depends on the phase content for multiphase coexistence. Thus, for the ceramics with R−O−T phase boundary, the $f$ can be modified as follows[34]:

$$f = F_R \times f_R + F_O \times f_O + F_T \times f_T \tag{8}$$

For the virgin state, the $f$ shows a decreasing trend from 50.5% to 39.2% with increasing $Sb^{5+}$ content (Fig. 3b). Compared with the virgin state, $f$ exhibits a relatively large improvement with an electric field of $3E_c$. The biggest $f$ is up to 81.8% for $x = 0.04$, which is probably ascribed to the high O phase content with spontaneous polarization of more available directions. After removal of the electric field, $f$ drops due to the depolarization process. Notably, the $f$ of the poled state is still larger than that of the virgin state, suggesting the contribution of the irreversible non-180° domain switching. The contribution of non-180° domain switching under various electric fields can be quantitatively analyzed according to the following equations[36]:

$$\Delta f_1 = f_{\text{in situ}} - f_{\text{poled}} \qquad (9)$$

$$\Delta f_2 = f_{\text{poled}} - f_{\text{virgin}} \qquad (10)$$

$$\Delta f_3 = f_{\text{in situ}} - f_{\text{virgin}} \qquad (11)$$

where $\Delta f_1$ and $\Delta f_2$ are the degree of reversible and irreversible non-180° domain switching, respectively, and $\Delta f_3$ is the degree of total non-180° domain switching. $\Delta f_1$, $\Delta f_2$, and $\Delta f_3$ decrease with increasing $Sb^{5+}$ content (Fig. 3c), corresponding with the variation of O phase content, implying reducing extrinsic contribution. In particular, the reduced irreversible non-180° domain switching is responsible for the decreased $S_{\text{rem}}$. The strain from reversible domain switching can be calculated based on the assumption that the change of strain is proportional to the poling texture change,

and it is determined by the following equation[36]:

$$S = \frac{\Delta f_1 \times S_{\text{neg}}}{\Delta f_2} \qquad (12)$$

where $S_{\text{neg}}$ is obtained from bipolar strain curves (Supplementary Fig. 5). It is obvious that the $S$ value first increases and then decreases with a level of 0.095–0.120% in the ceramics (Fig. 3d).

To further explore the contribution of irreversible domain switching, Rayleigh law based on phenomenological theory is also used to reveal the origin of strain[46–52]. The domain structure remains unchanged with the AC field cycle of $1/2E_c$ to $1/3E_c$ during the Raleigh region, leading to an increase linearly piezoelectric constant under the sub-switching conditions[50]. Their Rayleigh relationship can be expressed as follows[50]:

$$S(E) = (d_{\text{init}} + E_0)E \pm \alpha \frac{E_0^2 + E^2}{2} \qquad (13)$$

$$S(E_0) = (d_{\text{init}} + \alpha E_0)E_0 \qquad (14)$$

$$d(E_0) = d_{\text{init}} + \alpha E_0 \qquad (15)$$

where $S$ is strain value, $E$ is the driving electric field, $E_0$ is the maximum electric field, $d_{\text{init}}$ is the initial reversible piezoelectric response, $\alpha$ is the piezoelectric Rayleigh coefficient, and $\alpha E_0$ describes the irreversible piezoelectric response caused by the irreversible domain response. The $S$–$E$ loops are measured at an AC electric field with a maximum

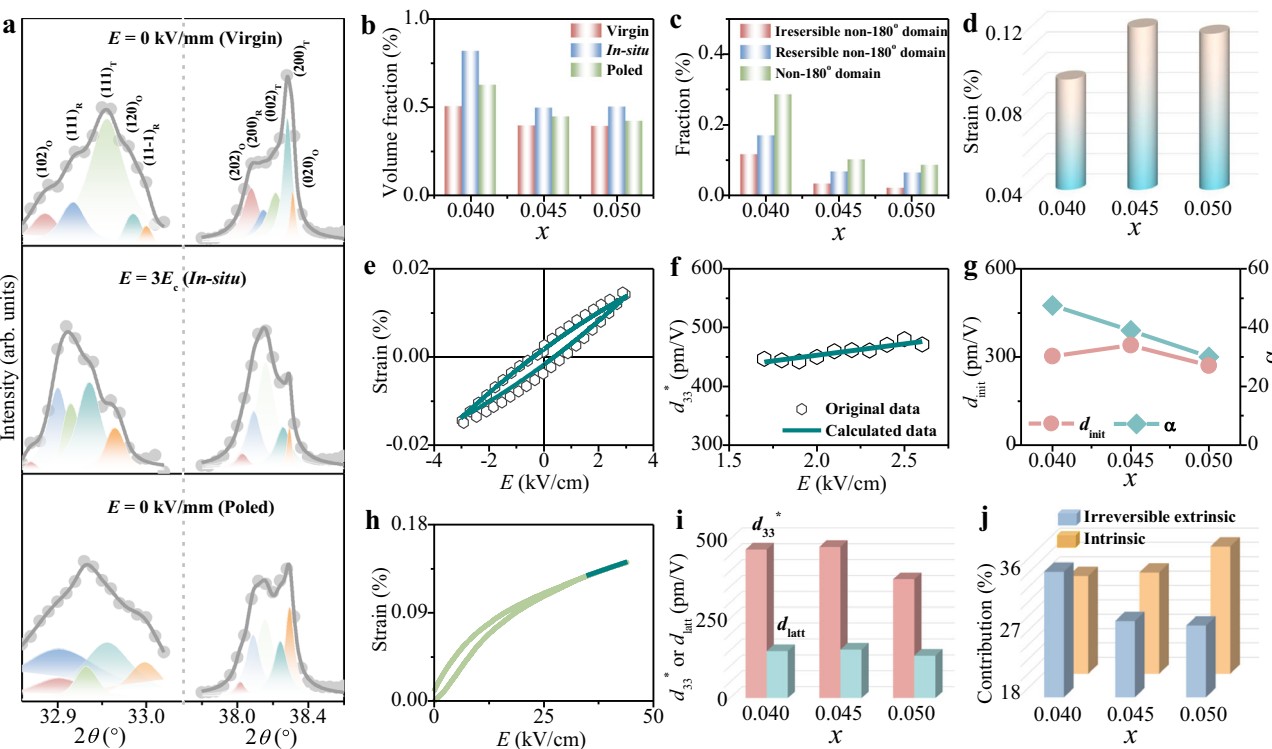

**Fig. 3 | Irreversible non-180° domain characterization with in situ synchrotron X-ray diffraction patterns and Rayleigh law. a** In situ synchrotron X-ray diffraction patterns of (111) and (200) reflections at 0 kV/cm, 3 times the coercive electric field ($3E_c$), and 0 kV/cm (poled) in the ceramics with $x = 0.045$. **b** Volume fraction of domains parallel to the poling direction. **c** Degree of domain switching under various electric fields. **d** Calculated reversible domain switching strain as a function of $x$. **e** Measured and fitted electric field-induced strain curves. **f** Piezoelectric coefficient with respect to the amplitude of the electric field. **g** Initial reversible piezoelectric response ($d_{\text{init}}$) and piezoelectric Rayleigh coefficient (α). **h** Unipolar strain curves at a high electric field. **i** Converse piezoelectric constant ($d_{33}^*$) and intrinsic piezoelectric coefficient ($d_{\text{latt}}$). **j** Irreversible extrinsic and intrinsic contributions.

amplitude of 3 kV/cm after poling at a high electric field. The fitting results suggest a high reliability for Rayleigh parameters in the ceramics (Fig. 3e and Supplementary Fig. 9a, d). The Rayleigh parameters of $d_{init}$ and α, corresponding to the reversible and irreversible piezoelectric response, can be calculated by linear fitting of measured $d_{33}^*$ as a function of electric field (Fig. 3f and Supplementary Fig. 9b, e), while $d_{33}^*$ is obtained according to the following equation[50]:

$$d^* = \frac{x_{p-p}}{2E_0} \qquad (16)$$

where $x_{p-p}$ is the difference between positive and negative strain values. The $d_{init}$ first increases and then decreases with increasing Sb$^{5+}$ (Fig. 3g). The enhancement when $x$ increases from 0.04 to 0.045 is ascribed to the modified R–O–T boundaries with flatter Gibbs free energy, while the decline for $x = 0.05$ results from the relative excessive diffusion behavior. The irreversible piezoelectric coefficient α decreases with the increasing content of Sb$^{5+}$, which is due to the decreased extrinsic contribution from irreversible domain evolution with decreasing O phase and strengthened diffusion behavior. Combining with the calculated $d_{init}$ and α, $d_{33}^*$ can be gained according to Eq. (15) with $E_0 = 3$ kV/cm (Fig. 3i). Then, the unipolar $S-E$ curve is measured at large electric fields for characterizing the intrinsic piezoelectric activity caused by the lattice distortion, while the domains are almost clamped and supply little contribution of domain switching or the motion of domain wall to the piezoelectric response. The slope for the linear part of $S-E$ loops is calculated as the intrinsic piezoelectric coefficient ($d_{latt}$) (Fig. 3h and Supplementary Fig. 9c, f). The $d_{latt}$ slightly increases first and then drops after adding Sb$^{5+}$ (Fig. 3i). In addition, the contribution of reversible domain wall motion ($d_{rever-DW}$) can be calculated as follows[50]:

$$d_{rever-DW} = d_{init} - d_{latt} \qquad (17)$$

Similar to the variation of $d_{latt}$ and $d_{init}$, the contribution of reversible domain wall motion also first increases and then decreases in the ceramics (Supplementary Fig. 10).

Thus, the proportion of intrinsic contribution ($C_{intri}$) from the lattice deformation can be calculated as[50]:

$$C_{intri} = \frac{d_{latt}}{d_{33}^*} \qquad (18)$$

The $C_{intri}$ increases with elevating Sb$^{5+}$, implying an enlarged contribution of lattice distortion (Fig. 3j), and the extrinsic contribution from the irreversible domain response can be evaluated as follows[50]:

$$C_{irrever} = \frac{d_{33}^* - d_{init}}{d_{33}^*} \qquad (19)$$

The decreased $C_{irrever}$ manifests that the weakened irreversible domain response is caused by the decreased amount of non-180° domain with decreasing O and increasing R/T phase, which is consistent with the results of in situ synchrotron XRD.

### Characterization of domain morphology

With optimized available strain from the modified irreversible non-180° domain, the domain morphology as well as switching behavior under bias is further explored[50]. The virgin domain structure is characterized as presented in Fig. 4a–c. Combining with the amplitude and phase images (Supplementary Fig. 11a–c), a refined domain is observed with increasing Sb$^{5+}$ content, due to the reduced O phase and strengthened diffusion instead of grain evolution that changes little (Supplementary Fig. 12). Then, the lithography process is adopted at

the regions under the bias of 2–8 V to analyze the domain switching behavior (Fig. 4d). Notably, although domain switching obvious happened at 4 V for $x = 0.04$ and 0.045 while little switching is found at 2 V, better switching is obtained for $x = 0.045$ than that for $x = 0.04$ (Fig. 4e, f and Supplementary Fig. 11d, e). The promoted domain switching is mainly generated by the modified phase content and diffusion behavior with a lowered energy barrier, which is also the origin of high piezoelectric performance (Supplementary Table 3). However, poor domain switching is observed for $x = 0.05$ due to diffused behavior which can make the domain recover easily after withdrawing bias (Fig. 4g and Supplementary Fig. 11f). Meanwhile, optimized piezoelectric response is also gained with the bias voltage-induced piezoresponse hysteresis and phase loops (Fig. 4h, i).

To reveal the domain morphology more accurately, TEM images are supplied as presented in Fig. 4j–l. A nano-sized stripe domain pattern is observed in all the ceramics along with uniformly distributed elements (Supplementary Fig. 13), conforming to the reported system with high performance from diffused R–O–T phase boundary[26,53–55]. The nano-sized stripe domain is further verified according to a regular fluctuated intensity at the regions marked by dotted lines (Fig. 4m–o), and the domain size decreases from 27 nm for $x = 0.04$ to 17 nm for $x = 0.045$, and 9 nm for $x = 0.05$ finally (Fig. 4p).

## Discussion

The contribution of the irreversible non-180° domain to available strain is closely associated with the phase content for multiphase coexistence. When the O phase decreases with increasing R and T phases, the irreversible non-180° domain gradually reduces because of the reduced non-180° domain wall, generating decreased $S_{rem}$ values. Meanwhile, large $S_{uni}$ is maintained well because of the existence of the R–O–T phase boundary which contains a strong converse piezoelectric effect and flexible domain response. Then, optimized available strain is realized with reduced O phase in the ceramics. By further decreasing the O phase and increasing the R and T phase ($x = 0.05$), not only the irreversible non-180° domain decrease, but also the diffusion is strengthened which signifies the enlarged random field that can accelerate the domain recovery after withdrawing the electric field, yielding to further increased $S_{uni}-S_{rem}$ along with lowered $S_{rem}$.

In addition, the irreversible non-180° domain evolution is also affected by the external field which induces structure transformation. Both $S_{rem}$ and $S_{uni}-S_{rem}$ gradually increase under elevating electric fields as $P_r$ and small signal $d_{33}$ increase rapidly when $E \leq 20$ kV/cm and almost stay high values when $E \geq 25$ kV/cm, which is due to the enhanced effective field applied on the irreversible non-180° domain with increasing electric field (Supplementary Fig. 14). Therefore, it can be noted that the higher electric field, the more sufficient domain switching and weaker domain reversion, manifesting that application of high electric field is beneficial of promoting available strain from increasing irreversible of the non-180° domain. With increasing temperature, the $S_{rem}$ and $S_{uni}-S_{rem}$ reduce owing to reduced irreversible non-180° domain wall from increased T phase and decreased R and O phases and lowered polarity as well as impeded polarization rotation (Supplementary Fig. 15). And small signal $d_{33}$ also decline sharply due to gradually vanished phase boundary, limited domain switching and reduced polarizability. Notably, strongly dropped $S_{rem}$ happens at the second cycle because most irreversible non-180° domains have been switched during the first cycle (Supplementary Fig. 16). Furthermore, little change is observed after three cycles, indicating little recovery of switched irreversible non-180° domains. Significantly, $S_{uni}-S_{rem}$ and $P_r$ exhibit slight fluctuation with increasing cycles, owing to sufficient domain switching for every cycle. The little change of $S_{neg}$ is observed when $n \geq 2$, which is attributed to the switched irreversible non-180° domain along the direction of the negative electric field after the first cycle. And, increasing $S_{rem}$ and $S_{uni}-S_{rem}$ are observed for the ceramics with increasing frequency, while $P_r$ presents a decreasing tendency

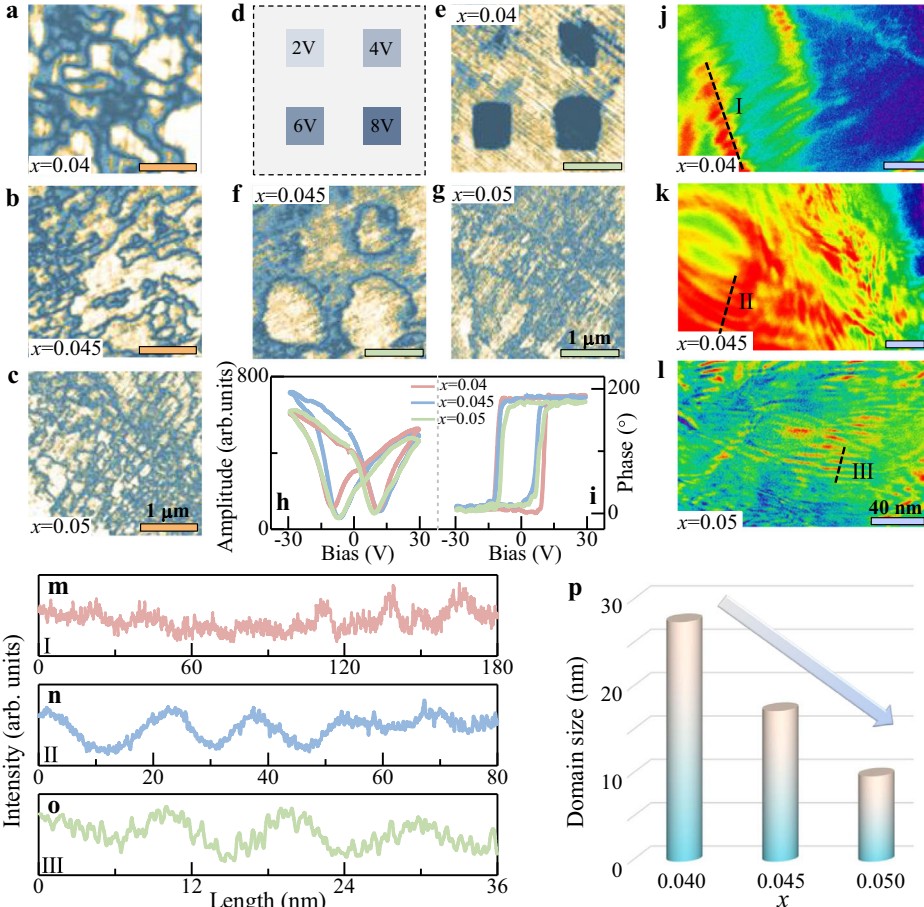

**Fig. 4 | Domain morphology and switching behavior.** Piezoresponse force microscopy (PFM) amplitude images of the ceramics with **a** $x = 0.04$, **b** $x = 0.045$, and **c** $x = 0.05$. Amplitude images after lithography process under bias: **d** schematic diagram of applying high voltages, **e** $x = 0.04$, **f** $x = 0.045$, and **g** $x = 0.05$. **h** Amplitude and **i** phase for switching spectroscopy piezoresponse force microscopy (SS-PFM) loops. Transmission electron microscopy (TEM) image of ceramics **j** $x = 0.04$, **k** $x = 0.045$, and **l** $x = 0.05$. Statistics of domain width of representative domains marked in the ceramics with **m** $x = 0.04$, **n** $x = 0.045$, and **o** $x = 0.05$. **p** Domain size as a function of $x$.

(Supplementary Fig. 17), indicating the decline of the irreversible non-180° domain. The higher the frequency, the poorer the sufficiency of domain switching. The lowered contribution of the irreversible non-180° domain appears accordingly.

In summary, the contribution of the non-180° domain to available strain is revealed based on the R−O−T phase boundary in KNN-based ceramics. Enlarged $S_{uni}−S_{rem}$ is achieved with significantly lowered $S_{rem}$ via reducing O and increasing R/T phase along with declined irreversible non-180° domain. According to the evolution of phase and domain structure, enhanced intrinsic and lowered extrinsic contributions are obtained to strain with increasing content of $Sb^{5+}$, attributing to the strengthened lattice distortion and reduced irreversible non-180° domain wall motion along with the improved random field. The mesoscopic manifestation is refined domain size and easy domain switching under external electric fields. Therefore, the modification of the irreversible non-180° domain is an effective method for designing high-performance functional materials for piezoelectric actuators.

## Methods
### Sample preparation
The ceramics with multiphase coexistence: $0.965K_{0.48}Na_{0.52}Nb_{1−x}$ $Sb_xO_3−0.02Bi_{0.5}Na_{0.5}ZrO_3−0.015CaZrO_3−0.4\%Fe_2O_3$ ($x = 0.035$, 0.04, 0.045, 0.05, and 0.055) were prepared via conventional solid-state method. The raw materials including $K_2CO_3$ (99.0%), $Na_2CO_3$ (99.8%), $Nb_2O_5$ (99.5%), $Sb_2O_3$ (99.99%), $Bi_2O_3$ (99.999%), $ZrO_2$ (99%), $CaCO_3$

(99%), and $Fe_2O_3$ (99%) produced by Sinopharm Chemical Reagent Co., Ltd., were ball milled for 24 h with both $ZrO_2$ ball media and alcohol after weighed according to the formula. The dried powders were calcined at 850 °C for 6 h with a heating rate of 3 °C min⁻¹, which were made into a disk with a diameter of ~10.0 mm and a thickness of ~1.0 mm under a pressure of 10 MPa with 8% PVA as a binder by the dry pressing process. After burning off the PVA of the disk, the specimens were sintered at 1070−1090 °C for 3 h with a heating rate of 3 °C min⁻¹ in air. Furthermore, the upper and under surface of the ceramics were covered by silver paste and then were sintered at 600 °C for 10 min. Finally, a direct current electric field of 3−4 kV/mm for 30 min was applied to the samples in a silicon oil bath.

### Structure characterizations
The XRD patterns were gained by XRD machine with Cu $k_\alpha$ radiation in the $\theta$-$2\theta$ scan mode (Bruker D8 Advanced XRD, Bruker AXS Inc., Madison, WI). The Materials Analysis Using Diffraction (MAUD) program was used to refine the XRD data to analyze the crystal structure parameters by selecting $KNbO_3$ (tetragonal $P4mm$, orthorhombic $Amm2$, and rhombohedral $R3m$) as the initial model. In situ, synchrotron XRD under 0 and $3E_c$ was performed using Source BL02U2 (photon energy 18 keV) beamline in Shanghai Synchrotron Radiation Facility. A thermally stimulated depolarization current was measured by an electrometer (Keithley 6517B, Keithley Instruments, Inc., Cleveland, OH). A conductive Pt−Ir-coated

cantilever PPP-NCHPt (Nanosensors, Switzerland) was applied to the Vertical Piezoresponse Force Microscopy (VPFM) with a commercial microscope (MFP-3D, Asylum Research, Goleta, CA). During the litho process, a negative voltage of 20 V was used first, and then a series of positive voltages (2, 4, 6, and 8 V) were employed in the oriented region. The triangular-wave high voltage was applied to obtain the local piezoresponse loops by the Switching Spectroscopy Piezo-response Force Microscopy (SS-PFM) mode. The argon-ion beam milling (Gatan PIPS 695, Gatan Inc., USA) with an operating voltage of 0.1–6 kV to reach electron transparency was used for Transmission Electron Microscopy (TEM) specimen with mechanical polishing to around 20 μm in thickness. A high-resolution transmission electron microscope (JEOL 2100F, JEOL, Japan) operated at 200 kV was used for the TEM. The morphology of the surface was carried out using the Field-emission Scanning Electron Microscope (FE-SEM) (JSM-7500, Japan).

### Electrical property measurements

The dielectric properties as a function of temperature in the range of −120–150 °C and 50–400 °C were gained by an *LCR* analyzer (E4980A, Keysight, USA). An impedance analyzer (Impedance Analyzer, PV70A, Beijing, China) was used to measure the planar electromechanical coupling factor ($k_p$) by a resonance-antiresonance method. A commercial Berlincourt-type $d_{33}$ meter (ZJ-3A, China) was applied to obtain the $d_{33}$ of the poled samples. A ferroelectric tester (aixACC TF Analyzer 2000E, Germany) was used for measuring the ferroelectric hysteresis ($P$–$E$) loops, $d_{33}$–$E$, and strain curves ($S$–$E$). The bipolar electric fields with amplitudes smaller than $1/2$ $E_c$ were applied to the samples according to the Rayleigh relationship under the sub-switching conditions. Then the unipolar $S$–$E$ curve under a large electric field with the slope of a linear part can be realized from the increasing applied electric fields.

## Data availability

The source data generated in this study are provided in the Source Data file. More relevant data sets generated during and/or analyzed during the current study are available from the first authors and corresponding authors on reasonable request. Source data are provided with this paper.

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

## Acknowledgements

K.W. acknowledges the support of the National Key Research and Development Program of China (No. 2020YFA0711700). H.T. acknowledges the support of the National Natural Science Foundation of China (No. 52202149), the Foundation of Sichuan Province Science and Technology Support Program (No. 2023NSFSC0968, 2021YJ0560, 22ZDYF3306, 2022NSFSC1970), and the Fundamental Research Funds for the Central Universities, Southwest Minzu University (No. ZYN2023110). S.-D.W acknowledges the support by the National Natural Science Foundation of China (No. 82202349), the Beijing Natural Science Foundation (No. L222066), and Peking University People's Hospital research and development funds (No. RDJP2022-44). B.H. acknowledges the support of the National Natural Science Foundation of China (No. 51972005, U21A2055). F.-Y. Zhu acknowledges the support of the National Natural Science Foundation of China (No. 52032005). We acknowledge W.-J.W. for the provision of PFM (MFP-3D, Asylum Research, Goleta, CA) experimental facilities, and the BL02U2 beamline of SSRF is acknowledged for the provision of experimental beamtime.

## Author contributions

B.W., H.T., S.-D.W., B.H., and K.W. conceived the idea of this work. B.W., L.Z., and J.-Q.F. prepared the KNN ceramics while Y.-T. Z., X.-L.S., Z.X., J.-T.L., and F.-Y.Z. performed the electrical and structural measurements and analyzed the data with J.M., H.T., and K.W. guided the project. J.M., S.-D.W., B.H., and K.W. provided financial and technical support for this work and provided indispensable suggestions together with Y.-X.L.

## Competing interests

The authors declare no competing interests.

## Additional information

[1]Sichuan Zoige Alpine Wetland Ecosystem National Observation and Research Station, Southwest Minzu University, Chengdu, P. R. China. [2]State Key Laboratory of New Ceramics and Fine Processing, School of Materials Science and Engineering, Tsinghua University, Beijing, P. R. China. [3]Sichuan Province Key Laboratory of Information Materials, Southwest Minzu University, Chengdu, P. R. China. [4]Musculoskeletal Tumor Center, Peking University People's Hospital, Beijing, P. R. China. [5]Shanghai Synchrotron Radiation Facility, Shanghai Advanced Research Institute, Chinese Academy of Sciences, Shanghai, P. R. China. [6]Department of Orthodontics, Peking University School and Hospital of Stomatology, Beijing, P. R. China. ✉e-mail: taohongscu@163.com; stonewang@bjmu.edu.cn; kqbinghan@bjmu.edu.cn; wang-ke@tsinghua.edu.cn

