## [Peer Review File · Nature Communications]

Contribution of irreversible non-180o domain to performance for multiphase coexisted potassium sodium niobate ceramicsREVIEWER COMMENTS

Reviewer #1 (Remarks to the Author):

Potassium sodium niobate (KNN) ceramics have been identified as one of the most promising lead-free piezoelectric candidates owing to their high Curie temperature and large piezoelectric properties. Surprisingly, different performance can be obtained in KNN-based ceramics even with the same phase-coexistence state, illustrating that the origin of high performance has not been sufficiently addressed apart from phase coexistence. This manuscript has thoroughly reported a crucial issue about irreversible non-180° domains' contribution in KNN-based ceramics. The authors applied experimentally to successfully illustrate the correlation between irreversible non-180° domains and electrical properties in high-performance KNN ceramics with phase coexistence. It is a meaningful work with systematic characterizations and reasonable discussions. I think that the result is interesting, and explains the main mechanism of the contribution to high performance in KNN ceramics. However, there are still some fatal weaknesses in the current manuscript that need to be clarified before publication:

1. The major findings in this work should be further discussed. It is interesting to note that ferroelectricity enables polycrystalline materials to exhibit piezoelectricity, but this ferroelectricity limits the level of practically achievable displacement bound by the difference between S_{pos} and S_{rem} for most actuator applications. S_{pos} involves intrinsic piezoelectric strain which mainly comes from crystal lattice distortion and extrinsic domain switching including irreversible and reversible non-180° domain switching. The reversible non-180° domains switch back, and the irreversible non-180° domains align along the electric field direction, resulting in S_{rem} . As the irreversible non-180° evolution has a great influence on the strain, the available strain ($S_{pos} - S_{rem}$) should be emphasized, which is important for most actuator applications. A more sufficient literature review is suggested.
2. As the R-O-T phase boundary is the basis of the discussion for domain evolution, the identification is not enough only with XRD and ϵ -T curves. More evidence should be adopted with accurate measurements. It is reported that temperature-dependent XRD and thermally stimulated depolarization current (TSDC) are effective methods for detecting phase structure, which may make the discussion more compelling.
3. Another important issue is that this current manuscript is not well-organized including the text and pictures. In particular, some data for assisting with discussion should be moved into Supplementary materials, such as relaxor characterization, et al.
4. The domain evolution is focused on and discussed in this paper. However, the domain structure characterization is lacking. More evidence, such as domain switching experiments, are recommended to be added.
5. It is insufficient as the authors only present ferroelectricity, small signal d_{33} and strain with different phase structures and domain evolution. Other important parameters, such as piezoelectric constant, should be also provided, which is closely related to irreversible non-180° domain switching and phase boundary.
6. In Figure 4(a)~(c), the legends are missed. Please check it carefully.
7. Some discussions need to be revised for more rigorous and logical language.
8. The measurement and characterization are too simple. Detailed information is required.
9. In Table S1, the values of TR-O seem to be not so accurate. The authors should provide more accurate values or adopt more accurate measurements.
10. There are some expression and grammar problems in the manuscript. The language needs to be polished carefully.

Reviewer #2 (Remarks to the Author):

Recently, the origin of high performance in KNN has been an interesting topic in the community. Generally speaking, the contribution can be broadly divided into the intrinsic and extrinsic contributions. The intrinsic part is related to lattice distortion while domain evolution belongs to the extrinsic contribution. Many recent works revealed the role of domain response in high-performance KNN, such as polar nanoregions (PNRs) (Energy Environ. Sci., 2018, 11, 3531-3539) and “nanoscale strain domains” (Adv.Mater., 2016, 28, 8519–8523). However, few studies focus on the domain type in KNN with high performance, especially the irreversible non-180° domain.

In this manuscript, the authors attempt to disentangle the question of “the effect of irreversible non-180° domain on performance in lead-free KNN ferroelectrics” and proceed quite successfully on the basis of a phenomenological description centrally involving the domain characteristic of high performance KNN with multiphase boundaries. It is clearly noticed that the basic underlying idea is the easily-neglected problems about the relationship between irreversible non-180° domain and performance in domain engineering, which is meaningful for high-precision lead-free actuator. The manuscript is well-organized and technically sound, and appropriate in supporting their claims. I would like to recommend its publication in Nature Communications after the authors have addressed the following concerns:

- (1) Since the explanation mainly depends on the domain configuration and its response, more direct evidence of the domain should be provided. TEM and PFM show a remarkable ability to observe and manipulate domain structure, which are widely adopted to detect the domain structure in piezoelectric materials. I suggest authors add the direct evidence of domain by TEM or PFM.
- (2) The evolution of domain vs. external fields would be a vital supplement to reveal the relationship between macroscopic performance and microscopic domain.
- (3) Why does the content of Sb change in such a narrow region (0.035~0.055)?
- (4) In Figs.4 and 6, the incomplete composition description might confuse readers.
- (5) The color contrast of data in many figures is insufficient, causing certain degree of reading difficulty. Please consider modifying these figures with better color palettes.
- (6) The manuscript contains too many figures, please consider summarizing into 4-5 figures.
- (7) Please show the ϵ -T curves for all the compositions in the supporting information.
- (8) There are some obvious grammatical mistakes throughout the paper, which should be corrected.

Reviewer #3 (Remarks to the Author):

The authors attempt to explain the contribution of the irreversible non-180° domain to performance in multi-phases-coexisting (i.e., rhombohedral, orthorhombic, and tetragonal) KNN materials. It is widely accepted that the non-180° domain plays a critical role in enhancing piezoelectric properties in piezoelectric materials, but the influence hasn't been fully explored in KNN-based ceramics. The authors try to clarify the change of the irreversible non-180° domain under different variables (e.g., electric fields, temperature and frequency), and the influence on the electrical properties. Such understanding would be beneficial to the design of high-performance lead-free piezoelectric ceramics. However, the experimental evidence of the evolution of the non-180° domain under electric field is lacking.

Additional measurements and careful analysis are needed to prove the contribution of domain structure.

Firstly, it has been noticed that piezoelectric response derives from intrinsic and extrinsic contribution, which mainly depend on lattice distortion and domain evolution including domain wall vibration and motion, domain switching, respectively. As the non-180° domain plays a role of extrinsic parts, the contribution to piezoelectric strain is not clear when the volume of the irreversible non-180° domain changes. The intrinsic and extrinsic contribution can be calculated to analyze the contribution of the non-180° domain to piezoelectricity.

Secondly, it is known that the contribution of the irreversible non-180° domain to piezoelectric strain depends on the external electric field. Thus, the degree of domain switching under different electric field amplitudes needs to be further investigated.

Meanwhile, clarify the difference between irreversible and reversible non-180° domain switching during piezoelectric strain measurement. Even though the ΔS_{neg} and ΔS_{rem} which come from irreversible non-180° domain is provided, more solid evidence is required.

Additionally, the manuscript is a bit lengthy and poorly organized, which might cause reading difficulty to the readers. The current form of this paper does not meet journal requirements.

The manuscript should be reorganized to highlight the core content of this study. Also, the Abstract and Introduction parts also need to be further optimized to improving the quality of manuscript.

Response Letter

(NCOMMS-23-52432)

Thank you very much for handing our manuscript and providing valuable comments. Here, we have revised the article actively and in detail with additional measurements and analysis according to your suggestion, to further improve the manuscript. We hope that these changes will fully address the concerns. All the reversion have been highlighted in red within the manuscript.

Reviewer#1: Potassium sodium niobate (KNN) ceramics have been identified as one of the most promising lead-free piezoelectric candidates owing to their high Curie temperature and large piezoelectric properties. Surprisingly, different performance can be obtained in KNN-based ceramics even with the same phase-coexistence state, illustrating that the origin of high performance has not been sufficiently addressed apart from phase coexistence. This manuscript has thoroughly reported a crucial issue about irreversible non-180° domains' contribution in KNN-based ceramics. The authors applied experimentally to successfully illustrate the correlation between irreversible non-180° domains and electrical properties in high-performance KNN ceramics with phase coexistence. It is a meaningful work with systematic characterizations and reasonable discussions. I think that the result is interesting, and explains the main mechanism of the contribution to high performance in KNN ceramics. However, there are still some fatal weaknesses in the current manuscript that need to be clarified before

publication:

Reply: Thank you very much for carefully reading and valuable comments about this work. According to your recommendations, we positively responded to your questions, as follows.

(1) The major findings in this work should be further discussed. It is interesting to note that ferroelectricity enables polycrystalline materials to exhibit piezoelectricity, but this ferroelectricity limits the level of practically achievable displacement bound by the difference between S_{pos} and S_{rem} for most actuator applications. S_{pos} involves intrinsic piezoelectric strain which mainly comes from crystal lattice distortion and extrinsic domain switching including irreversible and reversible non-180° domain switching. The reversible non-180° domains switch back, and the irreversible non-180° domains align along the electric field direction, resulting in S_{rem} . As the irreversible non-180° evolution has a great influence on the strain, the available strain ($S_{\text{pos}} - S_{\text{rem}}$) should be emphasized, which is important for most actuator applications. A more sufficient literature review is suggested.

Reply: Thank you for your reminding. Indeed, the available strain for the piezoelectric actuators is the difference between poling and remanent strain ($S_{\text{pol}} - S_{\text{rem}}$), which could be greatly affected by both intrinsic lattice distortion and extrinsic domain evolution. With the modification of irreversible non-180° domain, the S_{rem} changes as well, exhibiting influence on available strain. Thus, the change tendency of $S_{\text{pol}} - S_{\text{rem}}$ is added in this work. Significantly, $S_{\text{pol}} - S_{\text{rem}}$ gradually increases with decreasing contribution

of irreversible non-180° domain from reduced O and increased R/T phases, which can effectively reduce S_{rem} while S_{pol} could maintain because of the R-O-T phase boundary. For better discussion the important parameter as well as domain evolution, more sufficient literature review and systematical discussion have been conducted, as shown in the revised manuscript and below:

“a strategy of regulating the irreversible non-180° domain via phase engineering is introduced to optimize the available strain ($S_{uni}-S_{rem}$) in the lead-free potassium-sodium-niobate-based piezoelectric ceramic.”

“Generally, the actuating performance is assessed merely by the difference between poling strain (S_{pol}) and remanent strain (S_{rem}) (See Fig. 1a), which can be estimated via the following equation:

$$d_{33}^*=(S_{pol}-S_{rem})/E_{max}..... (1)$$

where the E_{max} is the maximum electric field and d_{33}^* is the converse piezoelectric coefficient. Obviously, the higher S_{pol} and the lower S_{rem} , the better actuating performance. Namely, the actuating performance can be modified by enhancing S_{pol} and/or restricting S_{rem} . S_{pol} can originate from multiple mechanisms, including the converse piezoelectric effect, non-180° domain-wall motion, electrostriction, and possible electric-field-induced phase transition. On the other hand, the S_{rem} is mainly generated by irreversible non-180° domain evolution which can keep the orientation after withdrawing the electric field. Compared to the complex mechanisms for S_{pol} , S_{rem} with a rather straightforward contribution of the irreversible non-180° domain might be easier to regulate. Modifying the irreversible non-180° domain could be an effective

approach to optimize available strain.”

“Significantly, a high available strain is realized with reduced irreversible non-180° domain and decreased domain size based on increased R/T and decreased O phase.”

“Improved piezoelectric and ferroelectric properties are achieved for $x=0.045$.

Meanwhile, strain evolution is further emphasized since it is a figure of merit for

piezoelectric actuators. From the unipolar strain curves (Fig. 2g), S_{rem} drops from 0.024%

to 0.019% when x increases from 0.04 to 0.05 along with little changed S_{uni} (~0.206%),

contributing to enhanced $S_{uni}-S_{rem}$, where the largest value of 0.187% is observed for

$x=0.05$ (Fig. 2h-2i). The high reversible strain achieved is favorable for the piezoelectric

actuating applications.”

Fig. 2 a Rietveld refinement of the ceramics with $x=0.04$, $x=0.045$, and $x=0.05$. b Phase

fraction. c R phase polarization direction. d O phase polarization direction. e T phase

polarization direction. f Temperature-dependent XRD of $x=0.045$. Evolution of g

unipolar strain, h S_{uni} , i S_{rem} , and j $S_{uni}-S_{rem}$.

(2) As the R-O-T phase boundary is the basis of the discussion for domain evolution, the identification is not enough only with XRD and ϵ_r - T curves. More evidence should be adopted with accurate measurements. It is reported that temperature-dependent XRD and thermally stimulated depolarization current (TSDC) are effective methods for detecting phase structure, which may make the discussion more compelling.

Reply: Thank you for your reminding. Besides XRD and ϵ_r - T curves, the temperature-dependent XRD patterns and j_{TSDC} have been measured to further analyze the phase structure. According to the temperature-dependent XRD patterns, the phase transition from R-O-T, to O-T, T and Cubic phases finally happens with increasing temperature. And, decreasing $T_{\text{O-T}}$ and T_{C} are obtained with increasing Sb^{5+} from j_{TSDC} , which is almost consistent with the results of ϵ_r - T curves. The detailed discussion has been presented in the revised manuscript, Supplementary materials, and below:

“Room-temperature phase transitions including $T_{\text{R-O}}$ and $T_{\text{O-T}}$ are observed for $x=0.035$ - 0.055 , indicating the R-O-T phase boundary (Supplementary Fig. 1c, Fig.2 and Table 1), which is verified by the Rietveld refinement (Fig. 2a, Supplementary Fig. 3, and Table 2).”

“The R-O-T phase boundary gradually changes to the O-T phase, T phase and cubic phase finally with increasing temperature, leading to the reduced non- 180° domain and weakened domain response (Fig. 2f).”

Supplementary Fig. 2 Normalized temperature dependence of depolarization current (j_{TSDC}) of the ceramics.

(3) Another important issue is that this current manuscript is not well-organized including the text and pictures. In particular, some data for assisting with discussion should be moved into Supplementary materials, such as relaxor characterization, et al.

Reply: Thank you for your reminding. All the text and pictures have been reorganized to emphasized the relationship between available strain and irreversible non-180° domain. The room-temperature XRD, ϵ_r - T curves, relaxor characterization, electrical properties, grain morphology haven been moved into Supplementary material. And the focuses including phase structure, available strain, *In-situ* X-ray diffraction, Rayleigh analysis and domain morphology have been carefully discussed in manuscript. The details are presented in the revised manuscript and supplementary material.

(4) The domain evolution is focused on and discussed in this paper. However, the domain structure characterization is lacking. More evidence, such as domain switching experiments, are recommended to be added.

Reply: Thanks for your reminding. The domain morphology has been added with PFM and TEM. Refined domain is observed with decreasing O and increasing R/T phases.

And the domain switching is promoted yielding from lowered energy barrier, which has been revealed by the lithography process under the bias of 2 V, 4 V, 6 V and 8 V.

The detailed discussion has been presented in the revised manuscript and below:

“With optimized available strain from the modified irreversible non-180° domain, the domain morphology as well as switching behavior under bias is further explored. The virgin domain structure is characterized as presented in Fig. 4a-c. Combining with the amplitude and phase images (Supplementary Fig. 11a-c), a refined domain is observed with increasing Sb^{5+} content, due to the reduced O phase and strengthened diffusion instead of grain evolution that changes little (Supplementary Fig. 12). Then, the lithography process is adopted at the regions under the bias of 2-8 V to analyze the domain switching behavior (Fig. 4d). Notably, although domain switching obvious happened at 4 V for $x=0.04$ and 0.045 while little switching is found at 2 V, better switching is obtained for $x=0.045$ than that for $x=0.04$ (Fig. 4e-f and Supplementary Fig. 11d-e). The promoted domain switching is mainly generated by the modified phase content and diffusion behavior with a lowered energy barrier, which is also the origin of high piezoelectric performance (Supplementary Table 3). However, poor domain switching is observed for $x=0.05$ due to diffused behavior which can make the domain recover easily after withdrawing bias (Fig. 4g and Supplementary Fig. 11f). Meanwhile, optimized piezoelectric response is also gained with the bias voltage-induced piezoresponse hysteresis and phase loops (Fig. 4h, i).

To reveal the domain morphology more accurately, TEM images are supplied as

presented in Fig. 4j-l. A nano-sized stripe domain pattern is observed in all the ceramics along with uniformly distributed elements (Supplementary Fig. 13), conforming to the reported system with high performance from diffused R-O-T phase boundary. The nano-sized stripe domain is further verified according to a regular fluctuated intensity at the regions marked by dotted lines (Fig. 4m-o), and the domain size decreases from 27 nm for $x=0.04$, to 17 nm for $x=0.045$ and 9 nm for $x=0.05$ finally (Fig. 4p).”

Fig. 4 PFM amplitude images of the ceramics with **a** $x=0.04$, **b** $x=0.045$, and **c** $x=0.05$.

Amplitude images after lithography process under bias: **d** schematic diagram of applying high voltages, **e** $x=0.04$, **f** $x=0.045$, and **g** $x=0.05$. **h** amplitude and **i** phase for

SS-PFM loops. TEM image of ceramics **j** $x=0.04$, **k** $x=0.045$, and **l** $x=0.05$. Statistics of domain width of representative domains marked in the ceramics with **m** $x=0.04$, **n** $x=0.045$, and **o** $x=0.05$. **p** Domain size as a function of x .

Supplementary Fig. 11 Phase images of the ceramics with **a** $x=0.04$, **b** $x=0.045$, and **c** $x=0.05$. Amplitude images after litho process under bias: **d** $x=0.04$, **e** $x=0.045$, and **f** $x=0.05$.

Supplementary Fig. 12 FE-SEM images of the ceramics: **a** $x=0.04$, **b** $x=0.045$, **c** $x=0.05$; statistics of grain size: **d** $x=0.04$, **e** $x=0.045$, **f** $x=0.05$.

Supplementary Fig. 13 Element mapping of the ceramic with $x=0.045$.

(5) It is insufficient as the authors only present ferroelectricity, small signal d_{33} and strain with different phase structures and domain evolution. Other important parameters, such as piezoelectric constant, should be also provided, which is closely related to irreversible non-180° domain switching and phase boundary.

Reply: Thanks for your reminding. The parameters including d_{33} , k_p , Q_m , ϵ_r , and $\tan\delta$ have been added, which are crucial to piezoelectric materials, as shown the revised manuscript, supplementary material and below:

“Especially, the variation of the non-180° domain could yield different performances (Supplementary Table 3 and Fig. 5). Improved piezoelectric and ferroelectric properties are achieved for $x=0.045$.”

Supplementary Table 3. Electric properties for the ceramics.

x	d_{33} (pC/N)	k_p	Q_m	ϵ_r	$\tan\delta$
-----	-----------------	-------	-------	--------------	--------------

0.04	495	0.482	56	2464	0.028
0.045	530	0.496	54	2749	0.024
0.05	505	0.495	59	2995	0.032

(6) In Figure 4(a)~(c), the legends are missed. Please check it carefully.

Reply: Thanks for your reminding. The details are carefully checked, and the missed legends have been added, as shown in the supplementary material:

Supplementary Fig. 5 **a** P - E loop. **b** Electric field-induced d_{33} curves. **c** Bipolar strain curves. **d** P_r and E_C . **e** d_{33} , **f** S_{pos} and S_{neg} for the ceramics.

(7) Some discussions need to be revised for more rigorous and logical language.

Reply: Thanks for your reminding. The description and discussion of the whole manuscript have been revised with more sufficient literature review and powerful evidence including *In-situ* X-ray diffraction, Rayleigh analysis and domain morphology as well as switching, to reveal the contribution of irreversible non-180° domain to

available strain in more depth. The detailed modification has been presented in the revised manuscript.

(8) The measurement and characterization are too simple. Detailed information is required.

Reply: Thank you for your reminding. More details have been provided for measurement and characterization, as presented in revised manuscript and below:

“Structure characterizations

The XRD patterns were gained by X-ray diffraction (XRD) machine with Cu k_{α} radiation in the θ - 2θ scan mode (Bruker D8 Advanced XRD, Bruker AXS Inc., Madison, WI). In-situ Synchrotron X-ray diffraction under 0 and $3E_c$ was performed using Source BL02U2 (photon energy 18 keV) beamline in Shanghai Synchrotron Radiation Facility. A thermally stimulated depolarization current measurement was conducted using an electrometer/high resistance meter (Keithley 6517B, Keithley Instruments, Inc., Cleveland, OH). Vertical piezoresponse force microscopy (VPFM) was conducted by a commercial microscope (MFP-3D, Asylum Research, Goleta, CA), applied to a conductive Pt-Ir-coated cantilever PPP-NCHPt (Nanosensors, Switzerland). During the litho process, a negative voltage of 20 V was used first, and then a series of positive voltages (2 V, 4 V, 6 V, 8 V) were employed in the oriented region. The local piezoresponse hysteresis loops were studied in the switching spectroscopy piezoresponse force microscopy (SS-PFM) mode, using triangular-wave high voltage. The transmission electron microscopy (TEM) specimen was prepared by mechanically

polishing to around 20 μm in thickness followed by argon-ion beam milling (Gatan PIPS 695, Gatan Inc., USA) with an operating voltage of 0.1–6 kV to reach electron transparency. The TEM was carried out using a high-resolution transmission electron microscope (JEOL 2100F, JEOL, Japan) operated at 200 kV. A field-emission scanning electron microscope (FE-SEM) (JSM-7500, Japan) was applied to characterize the morphology of the surface.

Electrical property measurements

The temperature-dependent dielectric properties at $-150\sim 200\text{ }^{\circ}\text{C}$ and $50\sim 450\text{ }^{\circ}\text{C}$ were measured by an LCR analyzer (E4980A, Keysight, U.S.A.). Their planar electromechanical coupling factor (k_p) was measured by a resonance-antiresonance method with an impedance analyzer (Impedance Analyzer, PV70A, Beijing, China). The d_{33} was measured with a commercial Berlincourt-type d_{33} meter (ZJ-3A, China) for the poled samples. The ferroelectric hysteresis (P - E) loops, d_{33} - E , and strain curves (S - E) were measured by a ferroelectric tester (aixACC TF Analyzer 2000 E, Germany). Following the Rayleigh relationship under the sub-switching conditions, bipolar electric fields with amplitudes smaller than $1/2 E_c$ were applied to the samples. Then increasing the applied electric fields, the unipolar S - E curve under a large electric field with the slope of a linear part can be obtained and estimated.”

(9) In Table S1, the values of T_{R-O} seem to be not so accurate. The authors should provide more accurate values or adopt more accurate measurements.

Response: Thanks for your reminding. It can be noted that R-O phase transition presents a diffusion behavior with a broad temperature region in KNN-based ceramics [*Chem. Soc. Rev.*, 2020, 49, 671, *Nat. Commun.*, 2021, 12], which is different from the sharp peaks of O-T and T-C phase transition during ϵ_r - T curves measurement. Therefore, the T_{R-O} is an approximate value, not an exact one.

(10) There are some expression and grammar problems in the manuscript. The language needs to be polished carefully.

Reply: Thanks for your reminding. The expression and grammar of the whole manuscript have been carefully checked and revised. Then, the manuscript has been also improved by an English native speaker. The details have been presented in the revised manuscript.

Reviewer #2: Recently, the origin of high performance in KNN has been an interesting topic in the community. Generally speaking, the contribution can be broadly divided into the intrinsic and extrinsic contributions. The intrinsic part is related to lattice distortion while domain evolution belongs to the extrinsic contribution. Many recent works revealed the role of domain response in high-performance KNN, such as polar nanoregions (PNRs) (*Energy Environ. Sci.*, 2018, 11, 3531-3539) and “nanoscale strain domains” (*Adv. Mater.*, 2016, 28, 8519–8523). However, few studies focus on the domain type in KNN with high performance, especially the irreversible non-180° domain. In this manuscript, the authors attempt to disentangle the question of “the effect of irreversible non-180° domain on performance in lead-free KNN ferroelectrics” and proceed quite successfully on the basis of a phenomenological description centrally

involving the domain characteristic of high performance KNN with multiphase boundaries. It is clearly noticed that the basic underlying idea is the easily-neglected problems about the relationship between irreversible non-180° domain and performance in domain engineering, which is meaningful for high-precision lead-free actuator. The manuscript is well-organized and technically sound, and appropriate in supporting their claims. I would like to recommend its publication in Nature Communications after the authors have addressed the following concerns:

Reply: We appreciate the reviewer for the interest and the valuable comments, which will definitely improve this paper.

(1) Since the explanation mainly depends on the domain configuration and its response, more direct evidence of the domain should be provided. TEM and PFM show a remarkable ability to observe and manipulate domain structure, which are widely adopted to detect the domain structure in piezoelectric materials. I suggest authors add the direct evidence of domain by TEM or PFM.

Reply: Thanks for your constructive suggestion. The domain configuration has been studied by PFM and TEM, which provide strong evidence for exploring the relationship between domain and performance in the ceramics. The related revision has been shown in the revised manuscript and below:

“With optimized available strain from the modified irreversible non-180° domain, the domain morphology as well as switching behavior under bias are further explored. The virgin domain structure is characterized as presented in Fig. 4a-c. Combining with the amplitude and phase images (Supplementary Fig. 11a-c), a refined domain is observed with increasing Sb^{5+} content, due to the reduced O phase and strengthened diffusion instead of grain evolution that changes little (Supplementary Fig. 12).”

“To reveal the domain morphology more accurately, TEM images are supplied as presented in Fig. 4j-l. A nano-sized stripe domain pattern is observed in all the ceramics

along with uniformly distributed elements (Supplementary Fig. 13), conforming to the reported system with high performance from diffused R-O-T phase boundary. The nano-sized stripe domain is further verified according to a regular fluctuated intensity at the regions marked by dotted lines (Fig. 4m-o), and the domain size decreases from 27 nm for $x=0.04$, to 17 nm for $x=0.045$ and 9 nm for $x=0.05$ finally (Fig. 4p).”

Fig. 4 PFM amplitude images of the ceramics with **a** $x=0.04$, **b** $x=0.045$, and **c** $x=0.05$. Amplitude images after lithography process under bias: **d** schematic diagram of applying high voltages, **e** $x=0.04$, **f** $x=0.045$, and **g** $x=0.05$. **h** amplitude and **i** phase for SS-PFM loops. TEM image of ceramics **j** $x=0.04$, **k** $x=0.045$, and **l** $x=0.05$. Statistics of domain width of representative domains marked in the ceramics with **m** $x=0.04$, **n**

$x=0.045$, and **o** $x=0.05$. **p** Domain size as a function of x .

Supplementary Fig. 11 Phase images of the ceramics with **a** $x=0.04$, **b** $x=0.045$, and **c** $x=0.05$. Amplitude images after litho process under bias: **d** $x=0.04$, **e** $x=0.045$, and **f** $x=0.05$.

Supplementary Fig. 13 Element mapping of the ceramic with $x=0.045$.

(2) The evolution of domain vs. external fields would be a vital supplement to reveal the relationship between macroscopic performance and microscopic domain.

Reply: Thanks for your suggestion. The domain vs. electric field has been carried out to reveal the relationship between macroscopic performance and microscopic domain

via PFM with litho mode, and the detailed revisions have been presented as follows:

“Then, the lithography process is adopted at the regions under the bias of 2-8 V to analyze the domain switching behavior (Fig. 4d). Notably, although domain switching obvious happened at 4 V for $x=0.04$ and 0.045 while little switching is found at 2 V, better switching is obtained for $x=0.045$ than that for $x=0.04$ (Fig. 4e-f and Supplementary Fig. 11d-e). The promoted domain switching is mainly generated by the modified phase content and diffusion behavior with a lowered energy barrier, which is also the origin of high piezoelectric performance (Supplementary Table 3). However, poor domain switching is observed for $x=0.05$ due to diffused behavior which can make the domain recover easily after withdrawing bias (Fig. 4g and Supplementary Fig. 11f). Meanwhile, optimized piezoelectric response is also gained with the bias voltage-induced piezoresponse hysteresis and phase loops (Fig. 4h, i).”

(3) Why does the content of Sb change in such a narrow region (0.035~0.055)?

Reply: This is a very good question. In this work, an appropriate R-O-T phase structure is expected to explore the effect on domain evolution from phase content. To obtain the R-O-T phase boundary, Sb is regarded as the most effective element to construct the phase structure due to the high efficiency of shifting phase transition temperature (T_{R-O} , T_{O-T}) in KNN ceramics. The experiment results show that the R and O phases coexist because of the high T_{O-T} for $x < 0.035$, while the R-O-T phase boundary with excessive diffusion behavior is obtained for $x > 0.055$. Therefore, a normal R-O-T phase structure can only gain in the Sb content from 0.035 to 0.055 in this material system.

(4) In Figs.4 and 6, the incomplete composition description might confuse readers.

Reply: Thanks for your kind comments. We have revised the description of Figs.4 (Supplementary Fig. 5 in revised version) and 6 (Supplementary Fig.14 in revised version) according to your kind reminding, and the detailed revisions are as follows:

Supplementary Fig. 5 **a** P - E loop of the ceramics. **b** Electric field-induced d_{33} curves. **c** Bipolar strain curves. **d** P_r and E_C . **e** d_{33} , **f** S_{pos} and S_{neg} for the ceramics: $x=0.04$, $x=0.045$, $x=0.05$.

Supplementary Fig. 14 **a** P - E loop. **b** Electric field induced d_{33} curves. **c** Bipolar strain curves. **d** Unipolar strain curves. **e** P_r , d_{33} , S_{neg} and S_{rem} . **f** $S_{uni}-S_{rem}$ under increasing electric field for $x=0.045$, measured at room temperature and 1 Hz.

(5) The color contrast of data in many figures is insufficient, causing certain degree of reading difficulty. Please consider modifying these figures with better color palettes.

Reply: Thanks for your kind advice. We have revised the color contrast based on your kind suggestion, and the detailed revisions have been presented in the revised manuscript.

(6) The manuscript contains too many figures, please consider summarizing into 4-5 figures.

Reply: This is a very good suggestion. We have summarized 4 figures in the manuscript, and others are listed in *Supplementary Materials*. The detailed 4 figures in the manuscript are as follows:

Fig. 1 a Schematic diagram of strain vs. domain evolution under applied electric field.

b Schematic diagram of the relationship between $S_{pol}-S_{rem}$ and irreversible non-180° domain.

Fig. 2 **a** Rietveld refinement of the ceramics with $x=0.04$, $x=0.045$, and $x=0.05$. **b** Phase fraction. **c** *R* phase polarization direction. **d** *O* phase polarization direction. **e** *T* phase polarization direction. **f** Temperature-dependent XRD of $x=0.045$. Evolution of **g** unipolar strain, **h** S_{uni} , **i** S_{rem} , and **j** $S_{uni}-S_{rem}$.

Fig. 3 **a** *In-situ* Synchrotron XRD patterns of (111) and (200) reflections at 0 kV/cm, $3E_c$, and 0 kV/cm (poled) in the ceramics with $x=0.045$. **b** Volume fraction of domains parallel to the poling direction. **c** Degree of domain switching under various electric

fields. **d** Calculated reversible domain switching strain as a function of x . **e** Measured and fitted electric field-induced strain curves. **f** Piezoelectric coefficient with respect to the amplitude of the electric field. **g** d_{init} and α . **h** Unipolar strain curves at a high electric field. **i** d_{33}^* and d_{latt} . **j** Irreversible extrinsic and intrinsic contributions.

Fig. 4 PFM amplitude images of the ceramics with **a** $x=0.04$, **b** $x=0.045$, and **c** $x=0.05$. Amplitude images after lithography process under bias: **d** schematic diagram of applying high voltages, **e** $x=0.04$, **f** $x=0.045$, and **g** $x=0.05$. **h** amplitude and **i** phase for SS-PFM loops. TEM image of ceramics **j** $x=0.04$, **k** $x=0.045$, and **l** $x=0.05$. Statistics of domain width of representative domains marked in the ceramics with **m** $x=0.04$, **n** $x=0.045$, and **o** $x=0.05$. **p** Domain size as a function of x .

(7) Please show the ϵ_r - T curves for all the compositions in the supporting information.

Reply: This is a very good suggestion. We have listed the ϵ_r - T curves in *Supplementary Materials*. The detailed revision are as follows:

Supplementary Fig. 1 XRD patterns of the ceramics with different x : **a** $2\theta=10\sim 80^\circ$, **b**

$2\theta=31\sim 33^\circ$ and $2\theta=45\sim 47^\circ$. **c** ϵ_r - T curves of the ceramics measured at $-120\sim 150^\circ\text{C}$.

Supplementary Fig. 4 **a** ϵ_r - T curves for the ceramics at $50\sim 400^\circ\text{C}$. **b** $\text{Ln}(1/\epsilon_r - 1/\epsilon_m)$ as

a function of $\text{Ln}(T - T_m)$. **c** $1000/\epsilon_r$ - T curves.

(8) There are some obvious grammatical mistakes throughout the paper, which should be corrected.

Reply: We have carefully checked the manuscript and polished the language throughout the whole article several times. Then, the manuscript has been also improved by an English native speaker. Please see our revised manuscript.

Reviewer #3: The authors attempt to explain the contribution of the irreversible non-180° domain to performance in multi-phases-coexisting (i.e., rhombohedral, orthorhombic, and tetragonal) KNN materials. It is widely accepted that the non-180° domain plays a critical role in enhancing piezoelectric properties in piezoelectric materials, but the influence hasn't been fully explored in KNN-based ceramics. The authors try to clarify the change of the irreversible non-180° domain under different variables (e.g., electric fields, temperature and frequency), and the influence on the electrical properties. Such understanding would be beneficial to the design of high-performance lead-free piezoelectric ceramics. However, the experimental evidence of the evolution of the non-180° domain under electric field is lacking. Additional measurements and careful analysis are needed to prove the contribution of domain structure.

Reply: We thank the reviewer for the positive comments and valuable suggestions.

(1) Firstly, it has been noticed that piezoelectric response derives from intrinsic and extrinsic contribution, which mainly depend on lattice distortion and domain evolution including domain wall vibration and motion, domain switching, respectively. As the non-180° domain plays a role of extrinsic parts, the contribution to piezoelectric strain is not clear when the volume of the irreversible non-180° domain changes. The intrinsic and extrinsic contribution can be calculated to analyze the contribution of the non-180° domain to piezoelectricity.

Reply: Thanks for your valuable advice. We have explored the contribution of non-180° domain to the performance through the Rayleigh law based on phenomenological

theory, and the detailed revision are as follows:

“To further explore the contribution of irreversible domain switching, Rayleigh law based on phenomenological theory is also used to reveal the origin of strain. The domain structure remains unchanged with the AC field cycle of $1/2E_c$ to $1/3E_c$ during the Raleigh region, leading to an increase linearly piezoelectric constant under the sub-switching conditions. Their Rayleigh relationship can be expressed as follows:

$$S(E) = (d_{\text{init}} + E_0)E \pm \alpha(E_0^2 - E^2)/2 \dots\dots\dots (13)$$

$$S(E_0) = (d_{\text{init}} + \alpha E_0) E_0 \dots\dots\dots (14)$$

$$d(E_0) = d_{\text{init}} + \alpha E_0 \dots\dots\dots (15)$$

where S is strain value, E is the driving electric field, E_0 is the maximum electric field, d_{init} is the initial reversible piezoelectric response, α is the piezoelectric Rayleigh coefficient, and αE_0 describes the irreversible piezoelectric response caused by the irreversible domain response. The S - E loops are measured at an AC electric field with a maximum amplitude of 3 kV/cm after poling at a high electric field. The fitting results suggest a high reliability for Rayleigh parameters in the ceramics (Fig.3e and Supplementary Fig. 9a and 9d). The Rayleigh parameters of d_{init} and α , corresponding to the reversible and irreversible piezoelectric response, can be calculated by linear fitting of measured d_{33}^* as a function of electric field (Fig. 3f, Supplementary Fig.9b and 9e), while d_{33}^* is obtained according to the following equation:

$$d^* = x_{\text{p-p}} / 2E_0 \dots\dots\dots (16)$$

where $x_{\text{p-p}}$ is the difference between positive and negative strain values. The d_{init} first increases and then decreases with increasing Sb^{5+} (Fig. 3g). The enhancement when x

increases from 0.04 to 0.045 is ascribed to the modified R-O-T boundaries with flatter Gibbs free energy, while the decline for $x=0.05$ results from the relative excessive diffusion behavior. The irreversible piezoelectric coefficient α decreases with the increasing content of Sb^{5+} , which is due to the decreased extrinsic contribution from irreversible domain evolution with decreasing O phase and strengthened diffusion behavior. Combining with the calculated d_{init} and α , d_{33}^* can be gained according to equation (15) with $E_0=3$ kV/cm (Fig. 3i). Then, the unipolar S - E curve is measured at large electric fields for characterizing the intrinsic piezoelectric activity caused by the lattice distortion, while the domains are almost clamped and supply little contribution of domain switching or the motion of domain wall to the piezoelectric response. The slope for the linear part of S - E loops is calculated as the intrinsic piezoelectric coefficient (d_{latt}) (Fig.3h, Supplementary Fig. 9c and 9f). The d_{latt} slightly increases first and then drops with adding Sb^{5+} (Fig. 3i). In addition, the contribution of reversible domain wall motion ($d_{\text{rever-DW}}$) can be calculated as follows:

$$d_{\text{rever-DW}} = d_{\text{init}} - d_{\text{latt}} \dots\dots\dots (17)$$

Similar to the variation of d_{latt} and d_{init} , the contribution of reversible domain wall motion also first increases and then decreases in the ceramics (Supplementary Fig. 10).

Thus, the proportion of intrinsic contribution (c_{intri}) from the lattice deformation can be calculated as:

$$c_{\text{intri}} = d_{\text{latt}}/d_{33}^* \dots\dots\dots (18)$$

The c_{intri} increases with elevating Sb^{5+} , implying an enlarged contribution of lattice distortion (Fig. 3j). And extrinsic contribution from the irreversible domain response

can be evaluated as follows:

$$c_{\text{irrever}} = (d_{33}^* - d_{\text{init}}) / d_{33}^* \dots \dots \dots (19)$$

The decreased c_{irrever} manifests that the weakened irreversible domain response is caused by the decreased amount of non-180° domain with decreasing O and increasing R/T phase, which is consistent with the results of In-situ Synchrotron XRD.”

(2) Secondly, it is known that the contribution of the irreversible non-180° domain to piezoelectric strain depends on the external electric field. Thus, the degree of domain switching under different electric field amplitudes needs to be further investigated. Meanwhile, clarify the difference between irreversible and reversible non-180° domain switching during piezoelectric strain measurement. Even though the S_{neg} and S_{rem} which come from irreversible non-180° domain is provided, more solid evidence is required.

Reply: This is a very good suggestion. *In-situ* X-ray diffraction (XRD) with the electric field is used to explore the contribution of irreversible non-180° domain to the available strain ($S_{\text{uni}} - S_{\text{rem}}$) in the samples, and the detailed revision are presented as follows:

“The electrostrain originates from both intrinsic lattice strain and extrinsic strain from the non-180° domain, where the latter often plays a major role. Thus, the evolution of domains upon applied electric field is explored. *In-situ* X-ray diffraction (XRD) has been recognized as a powerful approach to structure analysis during poling for piezoelectric materials. Notably, the shift of diffraction peak position and variation of peak intensity can reflect the piezoelectric lattice strain due to the change of the unit cell and non-180° domain switching, respectively. Of particular interest, the XRD patterns of (111) and (200) reflections are specifically investigated. The domain evolution under different electric fields (Virgin: 0 kV/mm → *In-situ*: $3E_c$ → Poled: 0

kV/mm) is revealed with well-fitted diffraction with R, O, and T phases (Fig. 3a and Supplementary Fig. 6). The volume fractions (F) of phases within the R-O-T multiphase coexistence can be approximately derived according to the following formula:

$$F_R = S_{111R} / [S_{111R} + S_{111O} + S_{111T}] \dots\dots\dots (2)$$

$$F_O = S_{111O} / [S_{111R} + S_{111O} + S_{111T}] \dots\dots\dots (3)$$

$$F_T = S_{111T} / [S_{111R} + S_{111O} + S_{111T}] \dots\dots\dots (4)$$

where S_{111R} , S_{111O} , and S_{111T} are the sums of peak intensities of (111) reflections for R, O, and T phases, respectively. The volume fractions of the phases are obtained from the characterization of the virgin state. Increasing R and T phases, as well as the decreasing O phase, are revealed due to increasing Sb^{5+} content (Supplementary Fig. 7), which conform to the result from the Rietveld refinement of XRD patterns. Importantly, a considerably enhanced volume fraction of non-180° domains parallel to the electric field direction is observed during the application of external electric field in all samples, which is evidenced by the changed peak intensity ratios between the 111_R/11-1_R, 202_O/020_O, and 002_T/200_T reflections for the R, O, and T phases, respectively. The volume fraction (f) of domains parallel to the electric field direction for R, O, and T phases can be expressed by the following equation:

$$f_R = I_{(111)R} / [I_{(111)R} + I_{(11-1)R}] \dots\dots\dots (5)$$

$$f_O = I_{(202)O} / [I_{(202)O} + I_{(020)O}] \dots\dots\dots (6)$$

$$f_T = I_{(002)T} / [I_{(002)T} + I_{(200)T}] \dots\dots\dots (7)$$

The f_R , f_O , and f_T first increase and then decrease in the ceramics within the applied electric field process from 0 kV/mm to $3E_c$, and subsequently to 0 kV/mm

(Supplementary Fig. 8), which can be ascribed to the reorientation of the non-180° domain under electric field. Significantly, the volume fraction of domains parallel to the electric field direction depends on the phase content for multiphase coexistence. Thus, for the ceramics with R-O-T phase boundary, the f can be modified as follows:

$$f = F_R * f_R + F_O * f_O + F_T * f_T \dots \dots \dots (8)$$

For the virgin state, the f shows a decreasing trend from 50.5% to 39.2% with increasing Sb^{5+} (Fig. 3b) content. Compared with the virgin state, f exhibits a relatively large improvement with an electric field of $3E_c$. The biggest f is up to 81.8% for $x=0.04$, which is probably ascribed to the high O phase content with spontaneous polarization of more available directions. After removal of the electric field, f drops due to the depolarization process. Notably, the f of the poled state is still larger than that of the virgin state, suggesting the contribution of the irreversible non-180° domain switching. The contribution of non-180° domain switching under various electric fields can be quantitatively analyzed according to the following equations:

$$\Delta f_1 = f_{in-situ} - f_{poled} \dots \dots \dots (9)$$

$$\Delta f_2 = f_{poled} - f_{virgin} \dots \dots \dots (10)$$

$$\Delta f_3 = f_{in-situ} - f_{virgin} \dots \dots \dots (11)$$

where Δf_1 and Δf_2 are the degree of reversible and irreversible non-180° domain switching respectively, and Δf_3 is the degree of total non-180° domain switching. Δf_1 , Δf_2 , and Δf_3 decrease with increasing Sb^{5+} (Fig. 3c) content, corresponding with the variation of O phase content, implying reducing extrinsic contribution. In particular, the reduced irreversible non-180° domain switching is responsible for the decreased S_{rem} .

The strain from reversible domain switching can be calculated based on the assumption that the change of strain is proportional to the poling texture change, and it is determined by the following equation:

$$S = \Delta f_1 * S_{neg} / \Delta f_2 \dots \dots \dots (12)$$

where S_{neg} is obtained from bipolar strain curves (Supplementary Fig. 5). It is obvious that the S value first increases and then decreases with a level of 0.095-0.120% in the ceramics (Fig. 3d). ”

Fig. 3 **a** *In-situ* Synchrotron XRD patterns of (111) and (200) reflections at 0 kV/cm, $3E_c$, and 0 kV/cm (poled) in the ceramics with $x=0.045$. **b** Volume fraction of domains parallel to the poling direction. **c** Degree of domain switching under various electric fields. **d** Calculated reversible domain switching strain as a function of x . **e** Measured and fitted electric field-induced strain curves. **f** Piezoelectric coefficient with respect to the amplitude of the electric field. **g** d_{mit} and α . **h** Unipolar strain curves at a high electric field. **i** d_{33}^* and d_{latt} . **j** Irreversible extrinsic and intrinsic contributions.

(3) Additionally, the manuscript is a bit lengthy and poorly organized, which might cause reading difficulty to the readers. The current form of this paper does not meet journal requirements. The manuscript should be reorganized to highlight the core content of this study. Also, the Abstract and Introduction parts also need to be further optimized to improving the quality of manuscript.

Reply: Thanks for your valuable suggestion. We have reorganized the whole manuscript based on the strategy of regulating irreversible non-180° domain via phase engineering to optimize the available strain. And we have also carefully checked the manuscript and polished the language, and then the manuscript has been also improved by an English native speaker. Please see our revised manuscript.

REVIEWERS' COMMENTS

Reviewer #1 (Remarks to the Author):

I appreciate that the authors have made substantial improvements to the manuscript, and the responses to comments also were addressed carefully. However, there are still some suggestions for this work, so I recommend that it should be published in Nature Communications after the following revisions:

- (1) The related Rietveld refinement information is lacking in this work, please add it in the "Structure characterizations" part.
- (2) In Supplementary Fig. 4 c, the " $\ln(1/\epsilon_r - 1/\epsilon_m)$ " and " $\ln(T - T_m)$ " should be revised as " $\ln(1/\epsilon_r - 1/\epsilon_m)$ " and " $\ln(T - T_m)$ " respectively.
- (3) The legend of Supplementary Fig. 4 b and c should be interchanged.
- (4) Please double-check the reference format.

Reviewer #2 (Remarks to the Author):

The revised manuscript by K. Wang et al is well-organized with quite interesting results. It deeply revealed the contribution of the irreversible non-180° domain-wall motion on the piezoelectric property of the KNN-based ceramics. This work not only broadened the knowledge of KNN-based systems but also offered a fancy strategy to study the piezoelectrics oxides.

In this version, in my opinion, all the concerns have been fully addressed and it can be published in its current form on Nature Communications preferentially.

Reviewer #3 (Remarks to the Author):

Authors have modified their manuscript in the current version of the manuscript and carefully made necessary response to the concerns or questions raised by the reviewer. It can thus be recommended for publication in its current form.

Response Letter

(NCOMMS-23-52432A)

Thank you very much for giving us the opportunity to address the remaining concerns of the reviewers before the manuscript is accepted for publication in your journal. We believe that the quality of the manuscript has been improved after those rounds of peer review, which would not be impossible without the careful reading and insightful suggestions of three professional reviewers. We have addressed the comments as follows and have prepared a revised version of the manuscript. All the reversions have been highlighted in red within the manuscript.

Reviewer#1: I appreciate that the authors have made substantial improvements to the manuscript, and the responses to comments also were addressed carefully. However, there are still some suggestions for this work, so I recommend that it should be published in Nature Communications after the following revisions:

Reply: Thank you very much for carefully reading and valuable comments about this work. According to your recommendations, we positively responded to your questions, as follows.

(1) The related Rietveld refinement information is lacking in this work, please add it in the “Structure characterizations” part.

Reply: Thank you for your reminding. We have added the related Rietveld refinement

information in the “Structure characterizations” part, and the detailed revisions are as follows:

“The Materials Analysis using Diffraction (MAUD) program were used to refine the XRD data to analyze the crystal structure parameters by selecting KNbO₃ (tetragonal *P4mm*, orthorhombic *Amm2*, and rhombohedral *R3m*) as the initial model.”

(2) In Supplementary Fig. 4 c, the “Ln(1/ε_r-1/ε_m)” and “Ln(T-T_m)” should be revised as “ln(1/ε_r-1/ε_m)” and “ln(T-T_m)” respectively.

Reply: Thank you for your reminding. We have revised the mistake according to your suggestion, and the detailed revisions are as follows:

Supplementary Fig. 4 a ϵ_r -T curves for the ceramics at 50~400 °C. **b** $1000/\epsilon_r$ -T curves.

c $\ln(1/\epsilon_r-1/\epsilon_m)$ as a function of $\ln(T-T_m)$.

(3) The legend of Supplementary Fig. 4 b and c should be interchanged.

Reply: Thank you for your reminding. We have revised the mistake according to your

suggestion, and the detailed revision is presented in revised manuscript.

(4) Please double-check the reference format.

Reply: Thanks for your reminding. We have carefully checked the reference format according to your suggestion, and the detailed revision is presented in revised manuscript.

Reviewer #2: The revised manuscript by K. Wang et al is well-organized with quite interesting results. It deeply revealed the contribution of the irreversible non-180° domain-wall motion on the piezoelectric property of the KNN-based ceramics. This work not only broadened the knowledge of KNN-based systems but also offered a fancy strategy to study the piezoelectrics oxides.

In this version, in my opinion, all the concerns have been fully addressed and it can be published in its current form on Nature Communications preferentially.

Reply: We thank the reviewer for the interest and support for this work. The constructive suggestions have helped us to improve this manuscript.

Reviewer #3: Authors have modified their manuscript in the current version of the manuscript and carefully made necessary response to the concerns or questions raised by the reviewer. It can thus be recommended for publication in its current form.

Reply: We thank the reviewer for the interest and support for this work. The helpful comments have helped us to improve this manuscript.